# Topographical Anatomy of the Rhesus Monkey (*Macaca mulatta*)—Part I: Thoracic Limb

**DOI:** 10.3390/vetsci10020164

**Published:** 2023-02-19

**Authors:** Christophe Casteleyn, Charlotte Gram, Jaco Bakker

**Affiliations:** 1Department of Morphology, Medical Imaging, Orthopedics, Physiotherapy and Nutrition, Faculty of Veterinary Medicine, Ghent University, Salisburylaan 133, 9820 Merelbeke, Belgium; 2Comparative Perinatal Development, Department of Veterinary Medicine, Faculty of Pharmaceutical, Biomedical and Veterinary Medicine, University of Antwerp, Universiteitsplein 1, 2610 Wilrijk, Belgium; 3Animal Science Department, Biomedical Primate Research Centre, Lange Kleiweg, 161, 2288 GJ Rijswijk, The Netherlands

**Keywords:** anatomy, topographical anatomy, rhesus monkey, thoracic limb

## Abstract

**Simple Summary:**

The rhesus monkey (*Macaca mulatta*) is one of the most investigated nonhuman primate species in biomedical research since its anatomy and physiology resemble that of humans. This manuscript fulfills the researcher’s and veterinarian’s need for detailed anatomical data on the rhesus monkey thoracic limb. Several rhesus monkey cadavers were dissected to study the anatomy of the muscular, circulatory, and peripheral nerve systems of the thoracic limb in relation to each other. The anatomical structures are textually described and illustrated by means of numerous detailed colored images.

**Abstract:**

Since the rhesus monkey (*Macaca mulatta*) is genetically closely related to man, it is generally accepted that its anatomy and physiology are largely similar to that of humans. Consequently, this non-human primate is most commonly used as a model in biomedical research. Not only the validation of the obtained research data, but also the welfare of the captive rhesus monkeys are subject to thorough anatomical knowledge of this species. Unfortunately, anatomical literature on the rhesus monkey is scarce, outdated, and hardly available at present. Furthermore, its anatomy is only illustrated by means of line drawings or black-and-white photographs. Thus, the aim of this study was to describe the anatomy of the thoracic limb of the rhesus monkey topographically, studying the various anatomical structures in relation to each other. In this manuscript, the anatomy of the thoracic limb is described per region. The structures that are visible on the different layers, from the superficial to the deepest layer, are described both in text and in numerous color images. As expected, the anatomy of the rhesus monkey is almost identical to human anatomy. However, some striking differences have been identified. This supports the necessity for an extensive publication on the anatomy of the rhesus monkey.

## 1. Introduction

The rhesus monkey (*Macaca mulatta*) is one of the most studied non-human primates [1]. This animal originates in the southern parts of Asia, but currently it can be found in research facilities worldwide [1,2]. Rhesus monkeys are commonly used in toxicity studies and their pivotal role in unraveling the mechanisms of various diseases should not be underestimated [3]. In addition, the development of many vaccines, e.g., against HIV, TBC and COVID-19, brings the rhesus monkey into play [4,5,6,7,8].

The demand of the rhesus monkey as an experimental model for human diseases finds its origin in the fact that the genomes of both species are nearly (93.5%) identical [9,10]. It is, therefore, not surprising that both species share many similarities [10]. However, if rhesus monkeys are used in studies that demand a comparable anatomy with humans, solid knowledge of its anatomy is essential for a correct interpretation of experimental results. Unfortunately, the literature on the anatomy of the rhesus monkey is rather insufficient.

Publications by Hartman and Straus Jr. [11], Berringer et al. [12], and Casteleyn and Bakker [13] may serve as important references regarding species specific anatomy. Unluckily, the circulatory system is documented less extensively. Therefore, the atlas by Swindler and Wood [14] on the comparative anatomy of the baboon (*Papio*), chimpanzee (*Pan*) and man (*Homo*) is invaluable. After all, both the rhesus monkey and the baboon belong to the tribe *Papionini*, and therefore show many similarities [2].

In addition to the rather brief descriptions of some anatomical systems, the publication by Hartman and Straus Jr. [11] only illustrates line drawings, whereas the atlas by Berringer et al. [12] illustrates black-and-white images. The latter approach lacks clarity, while the former fails to represent the authentic condition. The recent book chapter by Casteleyn and Bakker [13] shows many color images but does not describe the anatomy of the thoracic limb of the rhesus monkey in detail, as it claims to present the essentials for the biomedical researcher. Moreover, all these publications describe the anatomy per system and not per region (topographically). This may, for example, interfere with correct surgical intervention, e.g., when being confronted with wounds that must be tended to [15].

The arm (membrum thoracicum) is amongst the most complex and frequently used body parts. Malfunction of the hand (manus), wrist (carpus), lower arm (antebrachium), upper arm (brachium), and shoulder (cingulum membri thoracici) can be caused by injury, certain physical activities, or other health issues. As a result, surgical intervention in this body part is often requested. Detailed anatomical insight into the affected bones, joints, muscles, blood vessels, nerves, and lymph nodes is of the utmost importance to safeguard the welfare of the animals. Therefore, the aim of this study was to provide an overview of the topographical anatomy of the rhesus monkey thoracic limb.

## 2. Materials and Methods

### 2.1. Animals

A total of five rhesus monkey cadavers, three males and two females, were used in this study. The animals had been used in studies at the Biomedical Primate Research Centre (BPRC, Rijswijk, The Netherlands). Euthanasia was performed either when animals were found to be experiencing an unacceptable level of pain, stress, or distress (specified humane endpoints), or when the protocol required pathologic or histologic examination of organs or tissues, or when an animal reached the end of the study. Animals were fasted overnight and were sedated by intramuscular injection of ketamine hydrochloride (10 mg/kg ketamine 10%^®^, Alfasan Diergeneesmiddelen B.V., Woerden, The Netherlands) combined with medetomidine hydrochloride (0.05 mg/kg, Sedastart^®^, AST Farma B.V., Oudewater, The Netherlands). Subsequently, 70 mg/kg pentobarbital (Euthasol^®^ 20%; AST Farma B.V.) was injected intravenously (v. saphena parva). The cadavers were frozen at −18 °C and transported to the Laboratory of Morphology of Ghent University, Belgium, where they were thawed prior to anatomical investigation.

### 2.2. Dissection and Imaging

The left and right thoracic limbs of each rhesus monkey were subjected to anatomical dissection. As such, the presented results are based on ten thoracic limbs. Photographs, however, were only taken of the left thoracic limb, as the left lateral view is custom in most anatomy books and atlases. A Canon EOS 450D body (Canon Inc., Tokyo, Japan) combined with a Canon EF-S 18–200 mm f/3.5–5.6 IS lens (Canon Inc.) was used. Editing of the photographs was performed using GIMP 2.10.30 (gimp.org) and included cropping, adjusting the lighting, optimizing the color temperature, and providing a plain black background.

In order to facilitate the dissection of the blood vessels, latex injection was performed [16]. The arterial system of the thoracic limb was filled with red-colored latex (V-sure Eco Latex, Vosschemie Benelux, Belgium) by orthograde injection into the axillary artery. The superficial venous system was filled with blue-colored latex. The cephalic vein was injected in orthograde direction at the level of the carpus (wrist).

### 2.3. Anatomical Terminology

The terminology that is used throughout this article is based on the Nomina Anatomica Veterinaria [17]. Since this nomenclature focuses on domestic animals, it does not provide specific terms for the rhesus monkey. In the aforementioned publications on the anatomy of rhesus monkeys [11,12], human terminology was used. Unfortunately, not all human terms are applicable to or appropriate for the rhesus monkey since both species are not anatomically identical. This is confusing and prevents the comparison between different anatomical functionalities. For clarity, we opted to use veterinary nomenclature and have put any potential alternative or human term between square brackets. The latter terms were derived from the anatomy publications of Barone [18,19,20]—in which, besides the domestic mammals, the human is also described—or from a human anatomy atlas [21].

The Latin term has been used in the figure legends as well as the first time a structure is mentioned in the text. However, to increase readability, English terminology have been used to further detail the structures.

## 3. Results

This section presents the results of the anatomical dissections. First, the musculature that connects the thoracic limb to the body is discussed. Then the musculature of the shoulder, elbow, and carpal regions follows. The blood vessels, i.e., arteries and veins, and nerves that are in relation with the surrounding muscles are reviewed simultaneously. Per region, i.e., the shoulder region, the elbow region, the carpal region, and the hand, various views, such as a lateral, medial, ventral and dorsal view, are shown. Specifically for the hand, dorsal and palmar views are described.

### 3.1. Junctional Musculature Thorax-Thoracic Limb

#### 3.1.1. Dorsal Approach

After gently skinning the back, the neck, and the dorsal aspect of the junctional musculature between the thorax and the forelimb, the m. cutaneus trunci [m. panniculus carnosus], which covers the more caudal part of the m. latissimus dorsi, becomes visible (Figure 1A). The caudal part of the latissimus dorsi muscle is also thin and is located superficially. It originates in the last six thoracic vertebrae (Th6-Th12) and the thoracolumbar fascia to insert into the armpit, more specifically at the tuberositas teres major on the humerus. The caudoventral part of this muscle covers the craniodorsal aspect of the most superficial abdominal muscle, i.e., the m. obliquus externus abdominis. Cranial to the latissimus dorsi muscle, the triangular m. trapezius comes into sight. This muscle is separated by the spina scapulae into the caudal pars thoracica, which covers the craniodorsal portion of the latissimus dorsi muscle, and the cranial pars cervicalis. Just caudal to the scapular spine and cranial to the latissimus dorsi muscle, the opaque fascia m. infraspinati that covers the m. infraspinatus is visible. In addition, a glimpse of the m. teres major and m. rhomboideus thoracis can be observed.

Figure 1B portrays the middle layer after the removal of the cutaneus trunci muscle. It can now be observed that the latissimus dorsi muscle is composed of a cranial narrow but thick part, and a caudal wide but thin part. When both parts of the trapezius muscle are retracted, the entire m. rhomboideus thoracis [m. rhomboideus dorsi, m. rhomboideus major] and a large part of the m. rhomboideus cervicis [m. rhomboideus minor] can be discerned. These muscles connect the respective vertebral segments, hence their names, with the dorsal border of the shoulder blade (margo dorsalis scapulae). Cranial to the scapular spine, three previously hidden muscles appear. The m. supraspinatus, located in the fossa supraspinata, is, however, largely obscured by the m. atlantoscapularis cranialis [m. atlantoscapularis anterior/inferior], and the m. atlantoscapularis caudalis [m. atlantoscapularis posterior/superior]. The former muscle originates in the ventral aspect of the wing of the atlas (ala atlantis), runs in caudoventral direction parallel to the vertebral column, and inserts into the distal half of the spina scapulae, the acromion and the acromioclavicular joint. The latter muscle originates in the dorsal aspect of the ala atlantis, and runs in caudal direction dorsal to the cranial atlantoscapularis muscle to insert into the dorsal margin of the shoulder blade (margo dorsalis scapulae).

The cervical part of the trapezius muscle has been removed in Figure 1C, which shows the deepest layer of the junctional musculature. The third, most cranial part of the rhomboid muscle, i.e., the m. rhomboideus capitis, which runs towards the occiput, is now noticeable. The m. splenius capitis is located cranial to this muscle, and caudally partly obscured by it. This muscle, however, does not belong to the junctional musculature as it runs from the first three thoracic vertebrae towards the occiput. At the deepest level, the m. longissimus thoracis can be noticed immediately caudal to the m. rhomboideus thoracis, parallel to the vertebral column. The pale aponeurosis of the m. serratus dorsalis caudalis [m. serratus posterior inferior], which originates in the cervicothoracic fascia, is obvious in the dorsocaudal part of the thorax. Since the muscle fibers attach to the caudal ribs, it is not a junctional muscle, similar to the mm. intercostales externi that is located in between the ribs. On the other hand, the m. serratus ventralis thoracis [m. serratus anterior thoracis], which is cranially bordered by the armpit and the teres major muscle, and caudoventrally by the m. obliquus externus abdominis, conjoins the first nine ribs with the facies serrata on the medial sides of the shoulder blade.

**Figure 1 vetsci-10-00164-f001:**
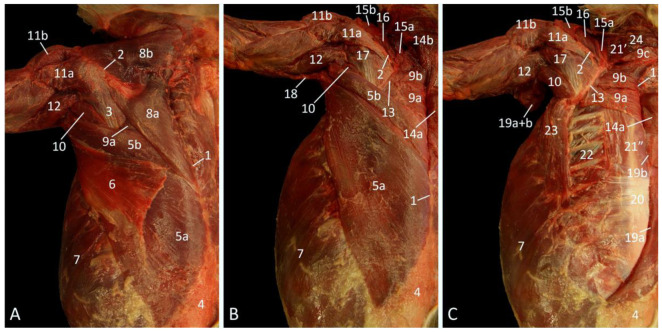
Dorsal view of the junctional musculature between the left thoracic limb and the thorax. (**A**): Superficial layer; (**B**): Middle layer, after the m. cutaneus trunci has been removed and the m. trapezius was retracted; (**C**): Deepest layer, after the removal of the m. latissimus dorsi and the m. trapezius pars cervicalis.

1: spinal processes, 2: spina scapulae, 3: fascia m. infraspinati, 4: fascia thoracolumbalis, 5a: caudal part of the m. latissimus dorsi, 5b: cranial part of the m. latissimus dorsi, 6: m. cutaneus trunci, 7: m. obliquus externus abdominis, 8a: m. trapezius pars thoracica, 8b m. trapezius pars cervicalis, 9a: m. rhomboideus thoracis, 9b: m. rhomboideus cervicis, 9c: m. rhomboideus capitis, 10: m. teres major, 11a: m. spinodeltoideus, 11b: m. acromiodeltoideus, 12: m. triceps brachii, 13: margo dorsalis scapulae, 14a: retracted m. trapezius pars thoracica, 14b: retracted m. trapezius pars cervicalis, 15a: m. atlantoscapularis caudalis, 15b: m. atlantoscapularis cranialis, 16: m. supraspinatus, 17: m. infraspinatus, 18: m. tensor fasciae antebrachii, 19a: stump of the caudal part of the m. latissimus dorsi, 19b: stump of the cranial part of the m. latissimus dorsi, 20: m. serratus dorsalis caudalis, 21′: cranial part of the m. longissimus thoracis, 21″: caudal part of the m. longissimus thoracis, 22: m. intercostalis externus, 23: m. serratus ventralis thoracis, 24: m. splenius capitis.

#### 3.1.2. Ventral Approach

The most substantial junctional muscles at the ventral side are the pectoral muscles. The m. pectoralis abdominalis [m. pectoralis superficialis pars abdominalis] can already be noticed in Figure 2A, which depicts the superficial layer. It is located immediately cranial to the m. obliquus externus abdominis, which aponeurosis forms the external sheath of the straight abdominal muscle that, together with the xiphoid process, functions as the site of origin of this muscle. The thin, triangular muscle that is located cranial to the m. pectoralis abdominalis, and in some specimens can be composed of two innominate parts, is the m. pectoralis transversus [m. pectoralis superficialis pars sternalis]. Both parts originate on the sternum and insert into the intertubercular groove of the humerus, albeit the caudal part originates more ventrally than the cranial part. The m. pectoralis descendens [m. pectoralis superficialis pars sternocapsularis] is the most cranially located pectoral muscle. It is the third component of the m. pectoralis superficialis [m. pectoralis major]. The m. pectoralis descendens finds its origin on the sternoclavicular joint and the manubrium. It inserts together with the cranial part of the m. pectoralis transversus and is cranially bordered by the m. cleidodeltoideus [m. deltoideus pars clavicularis, m. deltoideus anterior], which is the clavicular part of the m. deltoideus. This muscle has its origin on the entire clavicula and can be somewhat conjoined with the m. pectoralis superficialis. It will insert into the tuberositas deltoidea together with the other two parts of the deltoid muscle (see 3.2. Regio scapularis).

After the m. pectoralis transversus and the m. pectoralis descendens have been retracted, the deep m. pectoralis profundus [m. pectoralis minor, m. pectoralis ascendens] can be observed (Figure 2B). It finds its origin deep in the above-mentioned parts of the m. pectoralis superficialis on the sternum. The muscle fibers course in craniolateral direction to attach to the tuberculum majus humeri. This can also be seen in Figure 2C, in which not only the m. pectoralis profundus but also the m. pectoralis abdominalis have been retracted. The latter muscle clearly inserts in between the m. pectoralis profundus and the mm. pectorales transversus et descendens.

**Figure 2 vetsci-10-00164-f002:**
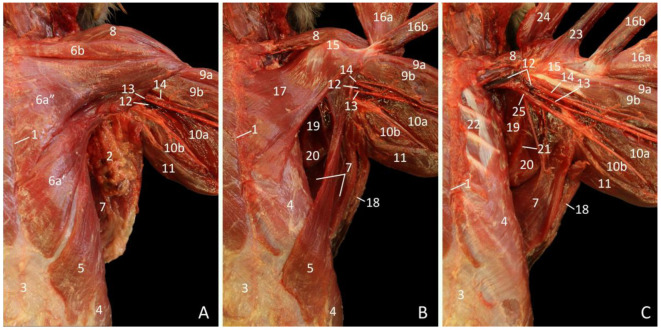
Dorsal view of the junctional musculature between the left thoracic limb and the thorax. The arterial system is filled with red latex. (**A**): Superficial layer; (**B**): Middle layer, after the removal of adipose tissue, the lnn. axillares accessorii and the caudal part of the m. pectoralis transversus, and after the cranial part of the de m. pectoralis transversus and the m. pectoralis descendens were retracted; (**C**): Deep layer, after retracting the m. pectoralis abdominalis and the m. pectoralis profundus.

1: sternum, 2: adipose tissue with lnn. axillares accessorii, 3: aponeurosis of the m. obliquus externus abdominis, 4: m. obliquus externus abdominis, 5: m. pectoralis abdominalis, 6a′: caudal part of the m. pectoralis transversus, 6a″: cranial part of the m. pectoralis transversus, 6b: m. pectoralis descendens, 7: m. latissimus dorsi, 8: m. cleidodeltoideus, 9a: m. biceps brachii caput longum, 9b: m. biceps brachii caput breve, 10a: m. triceps brachii caput mediale, 10b: m. triceps brachii caput longum, 11: m. tensor fasciae antebrachii, 12: v. axillaris, 13: a. axillaris, 14: n. medianus, 15: tuberculum minus humeri, 16a: retracted cranial parts of the m. pectoralis transversus and m. pectoralis descendens, 16b: retracted caudal part of the m. pectoralis transversus, 17: m. pectoralis profundus, 18: m. cutaneus trunci, 19: m. subscapularis, 20: m. teres major, 21: margo caudalis scapulae, 22: m. intercostalis externus, 23: retracted m. pectoralis abdominalis, 24: retracted m. pectoralis profundus, 25: n. ulnaris.

### 3.2. Regio Scapularis

The superficial shoulder musculature can be examined in Figure 3A. Distal to the scapular spine, the m. spinodeltoideus [m. deltoideus pars spinalis] and m. acromiodeltoideus [m. deltoideus pars acromialis] are visible. The origin of the former muscle is the scapular spine, while that of the latter muscle is the acromion. Both muscles attach together with the above-described m. cleidodeltoideus to the deltoid tuberosity of the humerus. The m. infraspinatus, which fills the fossa infraspinata of the shoulder blade, is concealed by the fascia m. infraspinati. It is largely visible as only its dorsal aspect is covered by the thoracic part of the trapezius muscle that attaches to the proximal third of the scapular spine. Since the cervical part of the trapezius muscle inserts into the entire length of the scapular spine, the m. supraspinatus, which fills the fossa infraspinata, is not yet exposed. The m. teres major is by now partly visible in between the infraspinous muscle, the latissimus dorsi muscle and the m. triceps brachii caput longum.

The infraspinatus muscle becomes exposed after retraction of the trapezius muscle and the removal of the fascia m. infraspinati, as shown in Figure 3B. Here, the supraspinatus muscle is still obscured by the mm. atlantoscapularis cranialis et caudalis [m. levator scapulae]. The three rhomboid muscles can be identified between the dorsal margin of the shoulder blade and the vertebral column.

When the cervical part of the trapezius muscle is completely removed (Figure 3C), the rhomboid muscles are fully visible. In addition, the m. longissimus thoracis, the m. splenius capitis, the external intercostal muscles, and the m. serratus ventralis thoracis come in view. By the subsequent removal of the latissimus dorsi muscle, the teres major muscle can be observed as a substantial, cylindrical muscle that is situated caudal to the shoulder blade. This muscle originates at the ventral angle and caudal border of the scapula and inserts medially into the proximal third of the humeral shaft.

Figure 3D shows that stage of the dissection in which the atlantoscapular muscles are removed. Consequently, the entire supraspinatus muscle is now observable. The rhomboid muscles are also detached, resulting in an inclusive view on the cranial part of the m. longissimus thoracis and on the m. splenius capitis.

**Figure 3 vetsci-10-00164-f003:**
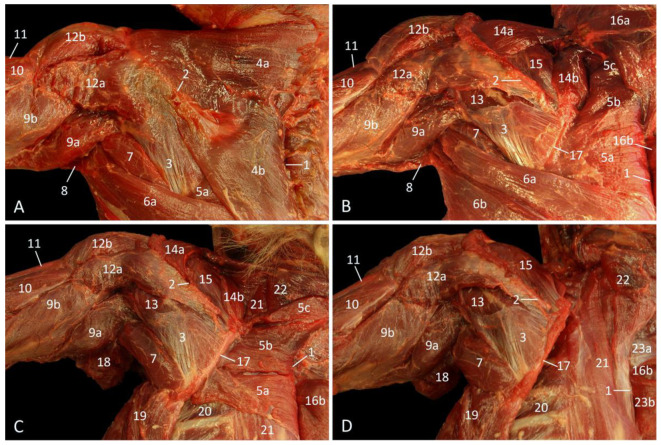
Dorsal view of the left shoulder. (**A**): Superficial layer, (**B**): Second layer, after the m. trapezius has been retracted, (**C**): Deep layer, after removing the m. latissimus dorsi, the m. tensor fasciae antebrachii and the m. trapezius pars cervicalis, (**D**): Deepest layer, after retracting the m. rhomboideus, and the mm. atlantoscapularis cranialis et caudalis.

1: spinal processes, 2: spina scapulae, 3: fascia m. infraspinati, 4a: m. trapezius pars cervicalis, 4b: m. trapezius pars thoracica, 5a: m. rhomboideus thoracis, 5b: m. rhomboideus cervicis, 5c: m. rhomboideus capitis, 6a: cranial part of the m. latissimus dorsi, 6b: caudal part of the m. latissimus dorsi, 7: m. teres major, 8: m. tensor fasciae antebrachii, 9a: m. triceps brachii caput longum, 9b: m. triceps brachii caput laterale, 10: m. brachialis, 11: m. biceps brachii caput longum, 12a: m. spinodeltoideus, 12b: m. acromiodeltoideus, 13: m. infraspinatus, 14a: m. atlantoscapularis cranialis, 14b: m. atlantoscapularis caudalis, 15: m. supraspinatus, 16a: retracted m. trapezius pars cervicalis, 16b: retracted m. trapezius pars thoracica, 17: margo dorsalis scapulae, 18: stump of the m. latissimus dorsi, 19: m. serratus ventralis thoracis, 20: m. intercostalis externus, 21: m. longissimus thoracis, 22: m. splenius capitis, 23a: retracted m. rhomboideus cervicis, 23b: retracted m. rhomboideus thoracis.

### 3.3. Regio Axillaris

In first instance, the axillary region is examined on a dorsolateral view, with the cadaver lying on its right lateral side (Figure 4). The shoulder blade has been detached from the thorax at its dorsal side. To this purpose, the latissimus dorsi and trapezius muscles were removed. Subsequently, the rhomboid muscles were transected, and the arm was rotated cranially.

The plexus brachialis, a network of nerves formed by the ventral branches of the cervical and thoracic nerves C5, C6, C7, C8 and Th1, is responsible for the innervation of the forelimb and a part of the thoracic wall. The n. suprascapularis originates from C5 and runs in ventrolateral direction to innervate both the supraspinatus and the infraspinatus muscles. From C6 leave the nn. subscapulares that penetrate the m. subscapularis that is located in the fossa subscapularis. This muscle is caudally delineated by the margo caudalis scapulae and the m. teres major. This muscle is innervated by the n. axillaris from C7. The n. thoracodorsalis from C8 can be seen in between the teres major muscle and the stump of the latissimus dorsi muscle. This nerve is responsible for the innervation of the latter muscle. The pectoral muscles receive nn. pectorales from Th1 that can be observed in the armpit during partial amputation of the forelimb.

As far as the hand and the lower and upper arm are concerned, the caudal nerve branches come into play. Associated with these are large arteries and veins. The massive nerve that runs in between the medial and long heads of the m. triceps, which it innervates by means of muscular branches, and finally leads towards the lateral side of the arm is the n. radialis. Adjoined to the caudal side of the n. radialis is the n. ulnaris that courses in the direction of the elbow. Before both nerves separate, the ulnar nerve provides the m. tensor fasciae antebrachii with a modest nerve branch. The nerve that runs in between the m. triceps brachii caput mediale and the m. biceps brachii caput breve is the n. medianus. It is joined by the a. brachialis, i.e., the principal artery of the arm, and the v. brachialis, i.e., the principal vein of the deep venous system of the arm.

**Figure 4 vetsci-10-00164-f004:**
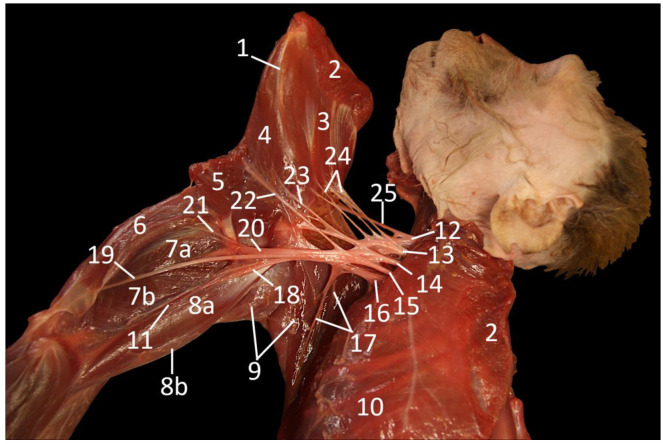
Dorsolateral view of the axillary region with emphasis on the plexus brachialis after partial amputation of the left forelimb that is rotated cranially. As such, a caudomedial view of the forelimb and a lateral view of the left thoracic wall are obtained.

1: margo caudalis scapulae, 2: stump of the m. rhomboideus, 3: m. subscapularis, 4: m. teres major, 5: stump of the m. latissimus dorsi, 6: m. tensor fasciae antebrachii, 7a: m. triceps brachii caput longum, 7b: m. triceps brachii caput mediale, 8a: m. biceps brachii caput breve, 8b: m. biceps brachii caput longum, 9: m. pectoralis transversus, 10: m. serratus ventralis thoracis, 11: a. brachialis, 12: C5 (fifth cervical nerve), 13: C6 (sixth cervical nerve), 14: C7 (seventh cervical nerve), 15: C8 (eighth cervical nerve), 16: Th1 (first thoracic nerve), 17: nn. pectorales, 18: n. medianus, 19: n. ulnaris, 20: n. radialis, 21: nerve branch of the n. ulnaris to the m. tensor fasciae antebrachii, 22: n. thoracodorsalis, 23: n. axillaris, 24: nn. subscapulares, 25: n. suprascapularis.

### 3.4. Regio Brachii

#### 3.4.1. Medial Approach

When the cadaver is positioned in dorsal recumbency, allowing for medial dissection of the upper arm, the pectoral muscles obscure the inspection of the armpit (Figure 5A). Caudally located in the axillary region, ventral to the latissimus dorsi muscle, are the axillary lymph nodes embedded in adipose tissue. The m. tensor fasciae antebrachii [m. dorsoepitrochlearis] is easily recognizable as a thin muscle that is stretched between the armpit and the elbow, covering the m. triceps brachii caput longum. The long head (caput longum) is the most caudal head of the m. triceps brachii, and also the most caudal muscle of the upper arm. Hence, it is observable from both the medial and the lateral side. The medial head (caput mediale) is located immediately cranial to the long head. Both heads insert into the olecranon of the ulna. The origins of the long and medial heads are the caudal border of the shoulder blade and the proximo-medial side of the humeral shaft, respectively. In the groove between both heads runs the n. ulnaris towards the extension angle of the elbow. The m. biceps brachii is located cranial to the medial head of the triceps muscle. In the groove between both muscles course the a. brachialis and the ramus distalis of the n. musculocutaneus towards the flexion angle of the elbow joint. The m. biceps brachii caput breve is in direct contact with this artery and nerve. The m. biceps brachii caput longum is positioned cranial to the short head. The long head (caput longum) and short head (caput breve) run from the supraglenoid tubercle and coracoid process of the shoulder blade, respectively, to the radial tuberosity of the radius. Their proximal aspects are overlaid by the cranial part of the m. pectoralis transversus.

The blood vessels and nerves of the upper arm can be examined in more detail after retracting the pectoral muscles and removing the adipose tissue with the embedded axillary lymph nodes (Figure 5B). The n. thoracodorsalis runs towards the m. latissimus dorsi across the m. subscapularis and the m. teres major. The n. medianus, with the a. brachialis parallel to it, can now be followed more proximally.

The blood vessels and nerves are elaborated in Figure 5C. The medianus and musculocutaneus nerves initially border the a. axillaris, the former at her caudal and the latter at her cranial side. After the ramus proximalis of the musculocutaneus nerve (ramus proximalis n. musculocutanei) has branched off to innervate the biceps brachii muscle, both nerves join each other. This nerve branch is accompanied by the a. circumflexa humeri cranialis that provides both bicipital heads with blood. Before the axillary artery becomes denominated by the term a. brachialis, the a. thoracodorsalis branches off to run to the latissimus dorsi muscle, together with the eponymous nerve. Along its trajectory, it laterally crosses the tenuous ulnar nerve and the heavy radial nerve. The latter nerve laterally crosses the a. collateralis ulnaris proximalis to ultimately disappear between the long and medial heads of the triceps brachii muscle. At the transition between the shoulder region and the upper arm, a second branch of the musculocutaneus nerve, i.e., the ramus distalis n. musculocutanei, is apparent. Like the proximal branch, it innervates the biceps brachii muscle. The median nerve continues its trajectory towards the flexion angle of the elbow adjacent to the brachial artery. The tuberculum minus humeri has come into sight by the removal of the pectoral muscles. Cranial to it is the cleidodeltoideus muscle. The m. coracobrachialis profundus and the m. coracobrachialis medialis are located caudal to the tuberculum minus humeri. The deep and the middle parts of the coracobrachialis muscle arise from the coracoid process on the shoulder blade. The shorter, former part inserts into the humeral neck, while the longer, latter part attaches more distally at the medial side of the humeral shaft.

The removal of the short head of the biceps brachii muscle exposes the humeral shaft and shows the a. bicipitalis that is responsible for the blood supply of the muscle to which her name refers (Figure 5D). The subsequent removal of the long head of the biceps brachii muscle together with some shoulder musculature exposes the m. brachialis (Figure 5E). This muscle, which is typically examined via a lateral approach, originates at the lateroproximal aspect of the humerus, then follows the brachial sulcus of that bone to insert into the medial coronoid process of the ulna. Due to the amputation of the cleidodeltoideus muscle, the tender m. subclavius can be seen onto the clavicle. The deep and medial coracobrachialis muscles are fully exposed by the excision of the biceps brachii muscle. By separating the long and medial heads of the triceps brachii muscle, the m. triceps brachii caput laterale can be observed from the medial side.

In Figure 5F, a large number of muscles has been taken away and the clavicle has been detached from the shoulder blade. The lateral head of the triceps brachii muscle, which finds its origin on the greater tuberosity of the humerus, remains on the upper arm together with the brachialis muscle. It is finally noteworthy to mention that the a. circumflexa humeri cranialis gives off a deep branch, i.e., the a. circumflexa humeri caudalis.

**Figure 5 vetsci-10-00164-f005:**
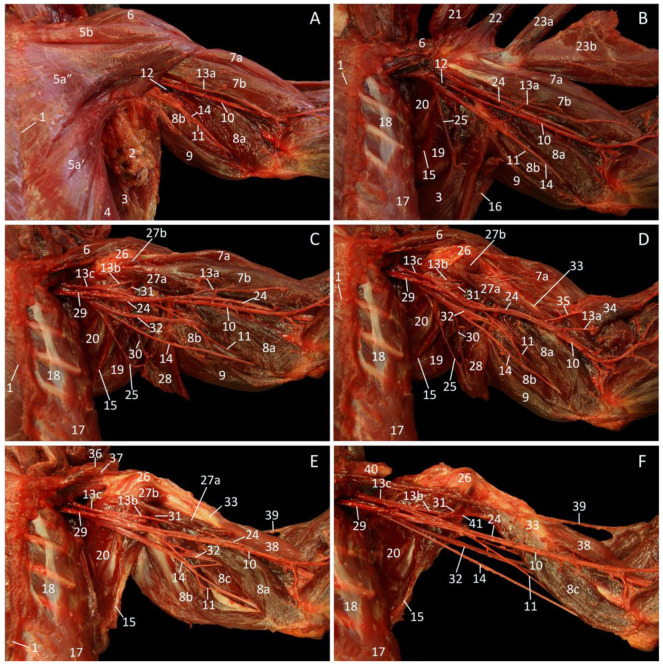
Medial view of the left armpit and upper arm; the arterial system is filled with red latex. (**A**): Superficial layer; (**B**): Second layer, after the m. pectoralis transversus, the m. pectoralis descendens, the m. pectoralis abdominalis, and the m. pectoralis profundus have been retracted, (**C**): Third layer, after detailed preparation of the blood vessels and nerves, and the removal of the mm. pectorales, the m. latissimus dorsi, and the m. cutaneus trunci; (**D**): Fourth layer, after removal of the m. biceps brachii caput breve; (**E**): Fifth layer, after the removal of the m. biceps brachii, the m. cleidodeltoideus, the m. teres major, the m. tensor fasciae antebrachii, and the stump of the m. latissimus dorsi; (**F**): Deepest layer, after the clavicle has been detached from the acromion and the m. triceps brachii caput longum and caput laterale, and the m. coracobrachialis medialis and profundus have been removed.

1: sternum, 2: adipose tissue with lnn. axillares accessorii, 3: m. latissimus dorsi, 4: m. pectoralis abdominalis, 5a′: caudal part of the m. pectoralis transversus, 5a”: cranial part of the m. pectoralis transversus, 5b: m. pectoralis descendens, 6: m. cleidodeltoideus, 7a: m. biceps brachii caput longum, 7b: m. biceps brachii caput breve, 8a: m. triceps brachii caput mediale, 8b: m. triceps brachii caput longum, 8c: m. triceps brachii caput laterale, 9: m. tensor fasciae antebrachii, 10: a. brachialis, 11: a. collateralis ulnaris proximalis, 12: v. axillaris, 13a: ramus distalis n. musculocutanei, 13b: ramus proximalis n. musculocutanei, 13c: n. musculocutaneus, 14: n. ulnaris, 15: margo dorsalis scapulae, 16: m. cutaneus trunci, 17: m. obliquus externus abdominis, 18: m. intercostalis externus, 19: m. teres major, 20: m. subscapularis, 21: retracted m. pectoralis profundus, 22: retracted m. pectoralis abdominalis, 23a: retracted caudal part of the de m. pectoralis transversus, 23b: retracted cranial part of the m. pectoralis transversus and the m. pectoralis descendens, 24: n. medianus, 25: n. thoracodorsalis, 26: tuberculum minus humeri, 27a: m. coracobrachialis medialis, 27b: m. coracobrachialis profundus, 28: stump of the m. latissimus dorsi, 29: a. axillaris, 30: a. thoracodorsalis, 31: a. circumflexa humeri cranialis, 32: n. radialis, 33: corpus humeri, 34: retracted m. biceps brachii caput breve, 35: a. bicipitalis, 36: clavicula, 37: m. subclavius, 38: m. brachialis, 39: v. cephalica, 40: detached clavicula, 41: a. circumflexa humeri caudalis.

#### 3.4.2. Lateral Approach

The musculature that is directly visible after skinning the cadaver is composed of the mm. deltoidei, the m. biceps brachii, and the m. triceps brachii (Figure 6A). As regards the deltoideus muscles, the spinodeltoideus muscle is located cranioproximally on the shoulder. The acromiodeltoideus muscle is situated caudal to the former muscle, in the middle of the shoulder region, while the m. triceps brachii caput longum is the most caudal muscle. The m. tensor fasciae antebrachii marginally covers its lateral aspect. The lateral head of the triceps brachii muscle is located cranial to and parallel with the long head. The distal portion of this head is in contact with the lower arm musculature, more specifically the m. extensor carpi radialis (longus et brevis) and the m. brachioradialis, which both find their origins on the distal aspect of the humerus. The brachialis muscle is situated between the brachioradialis muscle and the spinodeltoideus muscle that presents two bellies. The v. cephalica runs at the cranial border of the brachialis muscle (see 3.7. Vena cephalica).

The humerus becomes partly visible next to the removal of the deltoideus musculature (Figure 6B). In addition, the m. teres minor can now be identified as a small, cylindrical muscle immediately caudal to the m. infraspinatus. It has its origin at the caudodistal margin of the shoulder blade and the caudal aspect of the infraspinatus muscle. It inserts into the greater tubercle of the humerus, just caudal to the insertion of the aforementioned muscle.

Since most blood vessels and nerves are located at the medial side of the thoracic limb, they only become visible via a lateral approach, after the subsequent removal of the m. triceps brachii caput longum and the m. teres major (Figure 6C). The significant lateral head of the triceps brachii muscle is still in place. In the flexion angle of the shoulder joint, the brachial artery, median nerve, and ulnar nerve can be identified from cranial to caudal.

The brachialis muscle is completely isolated after the amputation of the lateral head of the triceps brachii muscle and the musculature of the lower arm (Figure 6D). This intervention additionally allows for the inspection of the medial head of the triceps brachii muscle from the lateral side. At this side, the radial nerve crosses the humerus and the brachialis muscle and continues in the direction of the flexion angle of the elbow. Caudal to this nerve, the ulnar nerve and the a. collateralis ulnaris proximalis run towards the caudal aspect of the elbow. The brachial artery and median nerve do not present any dissimilarities compared with Figure 6C. Inspection of the extension angle of the elbow reveals the m. anconeus lateralis [m. anconeus]. This small muscle arises distally on the humeral shaft and inserts proximally on the ulna.

**Figure 6 vetsci-10-00164-f006:**
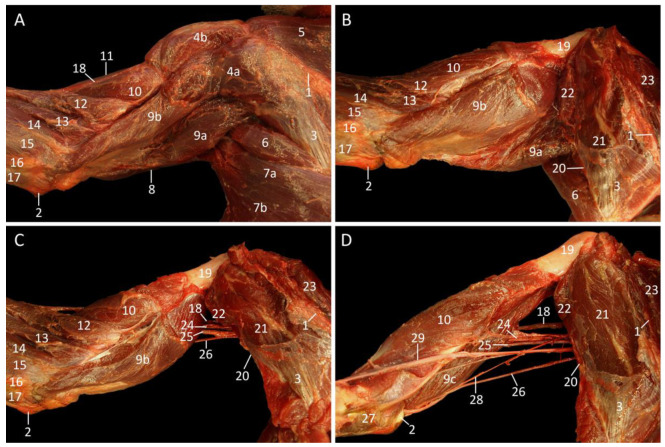
Lateral view of the left upper arm; the arterial system is filled with red latex. (**A**): Superficial layer; (**B**): Second layer, after the m. trapezius pars cervicalis, the m. biceps brachii caput longum, the m. spinodeltoideus, the m. acromiodeltoideus, and the m. tensor fasciae antebrachii have been taken away; (**C**): Third layer, with subsequent removal of the m. triceps brachii caput longum and the m. teres major; (**D**): Deepest layer, after amputation of the m. triceps brachii caput laterale, the m. brachioradialis, the m. extensor carpi radialis longus, the m. extensor carpi radialis brevis, the m. extensor digitorum communis, the m. extensor digitorum quarti et quinti proprius, and the m. extensor carpi ulnaris.

1: spina scapulae, 2: olecranon, 3: fascia m. infraspinati, 4a: m. acromiodeltoideus, 4b: m. spinodeltoideus, 5: m. trapezius pars cervicalis, 6: m. teres major, 7a: cranial part of the m. latissimus dorsi, 7b: caudal part of the m. latissimus dorsi, 8: m. tensor fasciae antebrachii, 9a: m. triceps brachii caput longum, 9b: m. triceps brachii caput laterale, 9c: m. triceps brachii caput mediale, 10: m. brachialis, 11: m. biceps brachii caput longum, 12: m. brachioradialis, 13: m. extensor carpi radialis longus, 14: m. extensor carpi radialis brevis, 15: m. extensor digitorum communis, 16: m. extensor digitorum quarti et quinti proprius, 17: m. extensor carpi ulnaris, 18: v. cephalica, 19: corpus humeri, 20: margo caudalis scapulae, 21: m. infraspinatus, 22: m. teres minor, 23: m. supraspinatus, 24: a. brachialis, 25: n. medianus, 26: n. ulnaris, 27: m. anconeus lateralis, 28: a. collateralis ulnaris proximalis, 29: n. radialis.

### 3.5. Regio Cubiti

The elbow region was dissected by a medial approach. Lateral views of this region can be found in the sections about the upper and lower arm. On the superficial layer (Figure 7A), the m. tensor fasciae antebrachii, which arises from the lower margin of the latissimus dorsi muscle and attaches to the antebrachial fascia and medial epicondyle of the humerus, forms the caudal border of the upper arm. This muscle is located caudal to the long head of the triceps brachii muscle. The medial head of the triceps brachii muscle, that is bilaterally bordered by blood vessels and nerves, is located cranial to the latter muscle, in the middle of the upper arm. In the groove between the long and medial heads of the triceps brachii muscle courses the n. ulnaris, accompanied by the a. collateralis ulnaris proximalis from the a. brachialis. More cranially, the n. medianus and the a. brachialis can be observed in the groove between the medial head of the triceps brachii muscle and the short head of the biceps brachii muscle. The brachial artery gives off the a. collateralis ulnaris distalis at the level of the elbow joint. Finally, the long head of the biceps brachii muscle can be found cranial to the short head and, as such, forms the cranial border of the upper arm.

After the a. collateralis ulnaris distalis has branched off, the a. brachialis divides into the a. radialis and the a. ulnaris. The latter artery is only visible on the deeper layers. In contrast, the former artery runs superficially on the forearm in between the m. extensor carpi radialis longus on the one hand side, and the m. pronator teres and m. flexor carpi radialis on the other hand side. The m. brachioradialis is located in the flexion angle of the elbow, cranial to the m. extensor carpi radialis longus. The brachioradialis muscle runs from the lateral humeral epicondyle to the distal aspect of the radius. The m. pronator teres is a triangular muscle that originates on the medial humeral epicondyle. It runs obliquely towards the middle third of the radius. Caudal to it lie the m. extensor carpi radialis (longus et brevis), the m. palmaris longus, and, finally, the m. flexor carpi ulnaris.

After the m. triceps brachii caput longum and caput mediale have been removed, the ulnar nerve can be followed caudally on the m. triceps brachii caput laterale (Figure 7B). This nerve runs at the level of the extension angle of the elbow, deep to the origins (i.e., the medial humeral epicondyle) of the m. flexor carpi ulnaris and the m. flexor digitorum superficialis [m. flexor digitorum sublimis]. The latter muscle has come into sight by the removal of the m. palmaris longus and the m. flexor carpi radialis, two muscles that also have their origins on the medial humeral epicondyle. At the level of the forearm, the ulnar nerve runs medial to the ulnar head (caput ulnare) of the m. flexor digitorum profundus, where she becomes flanked by the a. ulnaris.

The removal of both heads of the biceps brachii muscle exposes the m. brachialis that inserts into the medial coronoid process of the ulna (Figure 7C). When the m. flexor digitorum superficialis, the m. flexor carpi ulnaris, and the m. pronator teres are additionally taken away, the division of the brachial artery into the radial and ulnar arteries can be appreciated immediately proximal to the elbow joint. The radial artery continues her course at the cranial side of the forearm, along the m. extensor carpi radialis longus. The ulnar artery, on the other hand, presents a trajectory at the caudal side of the forearm, on the ulnar head of the m. flexor digitorum profundus that arises from the proximal half of the ulna. The first branch of the ulnar artery, i.e., the a. collateralis ulnaris distalis, can be noticed immediately proximal to the elbow joint. Just distal to the olecranon, the a. recurrens ulnaris is given off caudally, anastomosing with the proximal branch. Approximately at the same level but at the opposite side, the a. interossea communis branches off to accompany the median nerve. At the level of the elbow joint, this nerve gives off the ramus anastomoticus cum n. ulnari that is located parallel to the ulnar artery. As such, the median nerve is connected with the ulnar nerve. Cranial retraction of the cranially positioned m. extensor carpi radialis longus gives a view on the laterally located m. extensor carpi radialis brevis. Both heads have their origins on the lateral epicondylar crest of the humerus. The long head inserts into the base of the second metacarpal bone, while the short head attaches to the third metacarpal bone. In addition, the m. anconeus medialis [m. epitrochleoanconeus], which is a tiny muscle running from the medial humeral epicondyle to the olecranon, is visible.

**Figure 7 vetsci-10-00164-f007:**
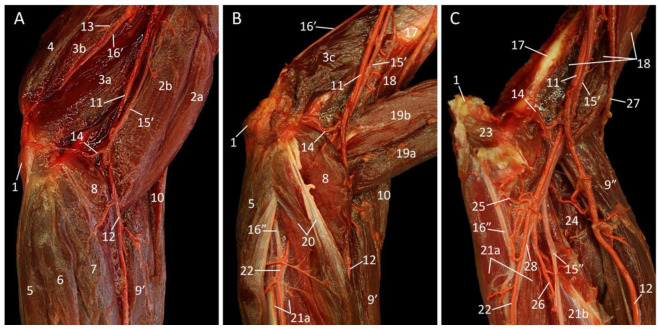
Medial view of the left elbow region; the arterial system is filled with red latex. (**A**): Superficial layer; (**B**): Deeper layer, after partial resection of the m. biceps brachii caput longum and caput breve, and the removal of the m. tensor fasciae antebrachii, the m. palmaris longus, the m. flexor carpi radialis, and the m. triceps brachii caput longum and caput laterale; (**C**): Deepest layer, after the removal of the m. triceps brachii caput laterale, the m. biceps brachii caput longum and caput breve, the m. flexor digitorum superficialis, the m. flexor carpi ulnaris, and the m. pronator teres.

1: olecranon, 2a: m. biceps brachii caput longum, 2b: m. biceps brachii caput breve, 3a: m. triceps brachii caput mediale, 3b: m. triceps brachii caput longum, 3c: m. triceps brachii caput laterale, 4: m. tensor fasciae antebrachii, 5: m. flexor carpi ulnaris, 6: m. palmaris longus, 7: m. flexor carpi radialis, 8: m. pronator teres, 9′: m. extensor carpi radialis longus, 9″: retracted m. extensor carpi radialis longus, 10: m. brachioradialis, 11: a. brachialis, 12: a. radialis, 13: a. collateralis ulnaris proximalis, 14: a. collateralis ulnaris distalis, 15′: brachial segment of the n. medianus, 15″: antebrachial segment of the n. medianus, 16′: brachial segment of the n. ulnaris, 16″: antebrachial segment of the n. ulnaris, 17: humerus, 18: m. brachialis, 19a: retracted m. biceps brachii caput longum, 19b: retracted m. biceps brachii caput breve, 20: retracted m. flexor digitorum superficialis, 21a: ulnar head of the m. flexor digitorum profundus, 21b: radial head of the m. flexor digitorum profundus, 22: a. ulnaris, 23: m. anconeus medialis, 24: m. extensor carpi radialis brevis, 25: a. recurrens ulnaris, 26: a. interossea communis, 27: v. cephalica, 28: ramus anastomoticus cum n. ulnari.

### 3.6. Regio Antebrachii

#### 3.6.1. Lateral Approach

The lateral side of the forearm is mainly characterized by the extensor musculature of the wrist and fingers. The m. brachioradialis, which runs from the most proximal aspect of the lateral humeral epicondylar crest towards the distal aspect of the radius, is located most cranially on a lateral view (Figure 8A). It cranially borders the m. extensor carpi radialis. The long head of this muscle (m. extensor carpi radialis longus) originates in between the m. brachioradialis and the short head of the extensor carpi radialis muscle (m. extensor carpi radialis brevis) on the lateral epicondylar crest of the humerus. The insertion sites of the long and short heads are the base of the second and third metacarpal bones, respectively. In the proximal half of the antebrachium, the m. brachioradialis cranially borders the muscular belly of the long head. In the distal half of the antebrachium, the brachioradialis muscle is located cranial to the tendon of the short head. The proximal half of the radius is laterally covered by the short head of the m. extensor carpi radialis and the m. extensor digitorum communis. The latter muscle arises from the lateral epicondyle of the humerus and inserts by means of four tendons into the distal phalanges of digits (fingers) II to V. The m. abductor digiti primi longus [m. abductor pollicis longus], which has its origin at the proximolateral aspect of the ulna and the craniodistal side of the radius is located just caudal to the radius, partially obscured by the m. extensor digitorum communis. It attaches to the proximal aspect of the metacarpal bone of the pollex. The m. extensor digitorum secundi (indicis) et tertii proprius, a slender muscle that is proximally laid over by the m. extensor digitorum quarti et quinti proprius, can be found more caudally. This muscle arises distal to the m. extensor digiti primi longus. At the level of the carpus, the tendon splits into two tendons, one to the proximal phalanx of the second digit and one for the third digit. The m. extensor digitorum quarti et quinti proprius has its origin at the lateral humeral epicondyle. The initial tendon also splits in two with a tendon inserting into the proximal phalanx of the fourth digit and one inserting into the middle phalanx of the fifth digit. Finally, the m. extensor carpi ulnaris, with its origin at the lateral epicondyle of the humerus and insertion at the base of the fifth metacarpal bone, is the caudal border of the forearm, albeit only in the proximal two thirds, since its tendon runs laterally over the m. flexor carpi ulnaris in the distal third.

When the m. extensor digitorum communis is retracted, the m. supinator, which is easily recognizable by its pale fascia, is obvious in the proximal half of the antebrachium (Figure 8B). It originates on the lateral humeral epicondyle and runs obliquely towards the proximal half of the radius.

After the m. extensor carpi radialis brevis has been retracted, two nerves come into sight (Figure 8C). The radial nerve is visible in the flexion angle of the elbow, where it gives off a ramus muscularis before it submerges underneath the supinator muscle. This muscular branch has a superficial trajectory along the m. extensor carpi radialis longus, which it innervates.

By the subsequent resection of the m. extensor carpi ulnaris, the ulna and the m. anconeus lateralis are exposed (Figure 8D). This muscle arises distally on the humeral shaft and inserts proximally on the ulna.

When the m. brachioradialis and the m. extensor carpi radialis longus are removed, the ramus muscularis of the radial nerve cannot be followed any longer. Instead, its ramus profundus can be seen in between the m. abductor digiti primi longus and the m. extensor digitorum secundi et tertii proprius (Figure 8E). It is a branch given off by the radial nerve cranial to the m. pronator teres that penetrates this muscle to re-emerge caudal from it.

Figure 8F shows the deepest layer. The membrana interossea antebrachii can be seen in between the radius and the ulna. On her medial aspect run the n. interosseus antebrachii and the a. interossea cranialis, parallel to the ulna.

**Figure 8 vetsci-10-00164-f008:**
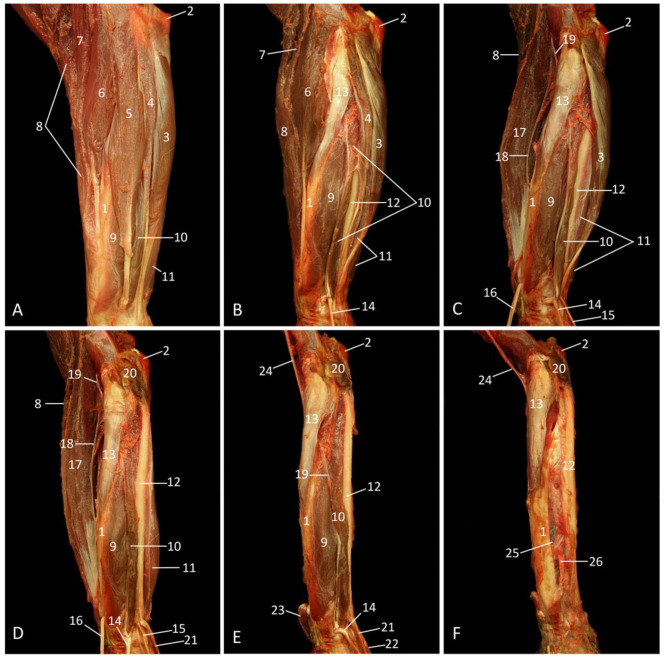
Lateral view of the left forearm; the arterial system is filled with red latex. (**A**): Superficial layer; (**B**): Second layer, after retraction of the m. extensor digitorum communis; (**C**): Third layer, after the removal of the m. extensor digitorum quarti et quinti and the m. extensor carpi radialis brevis; (**D**): Fourth layer, after excision of the m. extensor carpi ulnaris, (**E**): Fifth layer; after the m. brachioradialis, the m. extensor carpi radialis longus, and the m. flexor carpi ulnaris have been removed; (**F**): Deepest layer, after amputation of the m. extensor digitorum secundi et tertii proprius and the m. abductor digiti primi longus.

1: radius, 2: olecranon, 3: m. extensor carpi ulnaris, 4: m. extensor digitorum quarti et quinti proprius, 5: m. extensor digitorum communis, 6: m. extensor carpi radialis brevis, 7: m. extensor carpi radialis longus, 8: m. brachioradialis, 9: m. abductor digiti primi longus, 10: m. extensor digitorum secundi et tertii proprius, 11: m. flexor carpi ulnaris, 12: ulna, 13: m. supinator, 14: retracted tendon of the m. extensor digitorum communis, 15: retracted tendon of the m. extensor digitorum quarti et quinti proprius, 16: retracted tendon of the m. extensor carpi radialis brevis, 17: retracted m. extensor carpi radialis longus, 18: ramus muscularis n. radialis to the m. extensor carpi radialis longus, 19: n. radialis, 20: m. anconeus lateralis, 21: retracted tendon of the m. extensor carpi ulnaris: 22: retracted tendon of the m. flexor carpi ulnaris, 23: retracted tendon of the m. brachioradialis, 24: ramus profundus n. radialis, 25: membrana interossea antebrachii, 26: n. interosseus antebrachii and a. interossea cranialis

#### 3.6.2. Medial Approach

After skinning the forearm, the flexor musculature of the carpus and the digits can be studied (Figure 9A). The muscle that lies most cranially is the m. brachioradialis, which is caudally bordered by the m. extensor carpi radialis longus. Notice that these two muscles can be observed on both a lateral and a medial view. Caudal to the m. extensor carpi radialis longus runs the a. radialis. She is in contact with m. pronator teres and the m. flexor digitorum profundus in the proximal and distal half of the forearm, respectively. The latter muscle commences at the proximal half of the ulna (caput ulnare) and the upper two thirds of the radius (caput radiale). Five tendons arise, which are inserted into the palmar sides of the terminal phalanges of all five digits. Caudal to the m. pronator teres, thus proximally on the forearm, two muscle bellies can be seen that present their transitions into single tendons halfway up the forearm. The more cranial muscle is the m. flexor carpi radialis, which has its origin at the medial humeral condyle and its insertion at the base of the second metacarpal bone. The more caudally positioned muscle is the m. palmaris longus, which also originates at the medial humeral epicondyle. Its tendon runs over the m. flexor digitorum superficialis to finally present a distal aponeurosis (aponeurosis palmaris) that lies superficially at the palmar side of the hand. The most caudal muscle is the m. flexor carpi ulnaris. This muscle also arises from the medial humeral epicondyle. It attaches to the pisiform carpal bone.

When the two most superficial muscles, i.e., the m. palmaris longus and the m. flexor carpi radialis, are removed or retracted, the entire m. flexor digitorum superficialis is exposed (Figure 9B). The m. flexor digitorum superficialis originates at the medial epicondyle of the humerus. Its four tendons insert into the base of the second phalanx of digits II to V.

In Figure 9C, the m. flexor digitorum superficialis is largely removed, revealing some blood vessels and veins. The ulnar nerve is located at the caudal side of the antebrachium, in between the m. flexor carpi ulnaris and the m. flexor digitorum profundus, is accompanied by the ulnar artery from halfway to the antebrachium and onwards. Cranial to both structures run the median nerve, artery, and vein parallel to each other. They are initially positioned in between the m. flexor digitorum superficialis and the m. pronator, and finally run over the m. flexor digitorum profundus.

Once the entire m. flexor digitorum superficialis is removed, the extent of the m. flexor digitorum profundus is remarkable (Figure 9D). This muscle covers both the radius and the ulna. Major blood vessels and nerves are now visible as they proceed onto this muscle. It was earlier described that, proximal to the elbow, the brachial artery splits into the radial and ulnar arteries, located at the cranial and caudal sides of the antebrachium, respectively. It can now be observed that the radial artery gives off the a. interossea communis just distal to the elbow. From this artery two branches successively emerge. This layer allows for the visualization of the above-mentioned median artery that halfway to the antebrachium gets company from the median nerve. The ramus anastomoticus cum n. ulnari, which branches off from the median nerve just distal to the elbow joint to run in caudodistal direction towards the ulnar nerve, is also visible.

The second branch of the radial artery, i.e., the a. interossea cranialis [a. interossea anterior, a. interossea palmaris], comes into sight after resection of the m. brachioradialis, the m. extensor carpi radialis longus, the m. flexor carpi ulnaris, and the m. flexor digitorum profundus (Figure 9E). It runs in between the radius and ulna, onto the membrana interossea antebrachii, with the n. interosseus antebrachii [n. interosseus antebrachii volaris] next to it. Both structures become distally obscured by the m. pronator quadratus, in contrast to the median nerve that stays at the medial side of this muscle. The radial nerve can be seen at the cranial side of the antebrachium, in between the brachial artery and the radius. At the caudal side of the antebrachium, the m. extensor carpi ulnaris can be identified. Finally, the m. anconeus medialis can be noticed.

The deepest layer of Figure 9F is devoid of most muscles, blood vessels, and nerves. Only the radius, ulna and the intermediary structures remain. These include the above-described membrana interossea antebrachii with the a. interossea cranialis and the n. interosseus antebrachii at her medial side. The m. pronator quadratus is a rectangular muscle located in the distal fourth of the antebrachium, running from the proximal aspect of the ulna to the distal aspect of the radius.

**Figure 9 vetsci-10-00164-f009:**
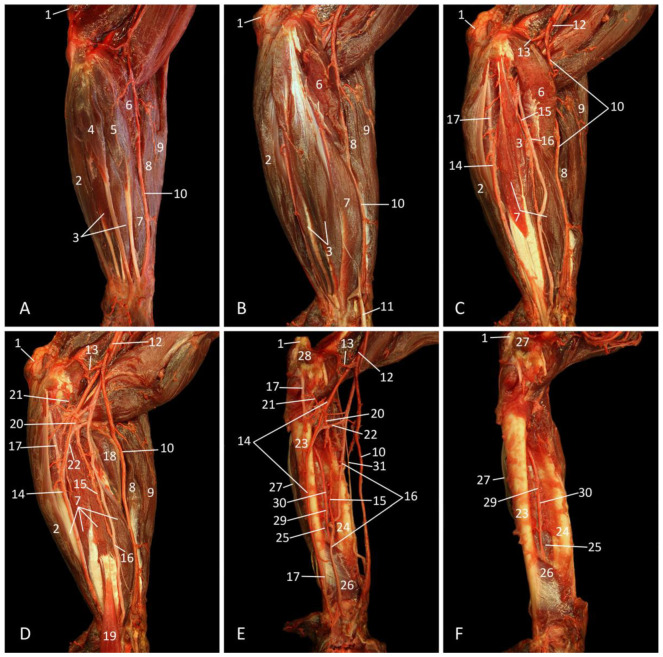
Medial view of the left forearm; the arterial system is filled with red latex. (**A**): Superficial layer; (**B**): Second layer, after the m. flexor carpi radialis has been retracted and the m. palmaris longus was removed; (**C**): Third layer, after partial resection of the m. flexor digitorum superficialis; (**D**): Fourth layer, after the total resection of the m. flexor digitorum superficialis and the m. pronator teres; (**E**): Fifth layer, after amputation of the m. brachioradialis, the m. extensor carpi radialis longus, the m. flexor carpi ulnaris, and the m. flexor digitorum profundus; (**F**): Deepest layer, after the removal of the majority of blood vessels and nerves.

1: olecranon, 2: m. flexor carpi ulnaris, 3: m. flexor digitorum superficialis, 4: m. palmaris longus, 5: m. flexor carpi radialis, 6: m. pronator teres, 7: m. flexor digitorum profundus, 8: m. extensor carpi radialis longus, 9: m. brachioradialis, 10: a. radialis, 11: retracted tendon of the m. flexor carpi radialis, 12: a. brachialis, 13: a. collateralis ulnaris distalis, 14: a. ulnaris, 15: a. mediana, 16: n. medianus, 17: n. ulnaris, 18: m. extensor carpi radialis brevis, 19: retracted tendon of the m. flexor digitorum superficialis, 20: a. interossea communis, 21: a. recurrens ulnaris, 22: ramus anastomoticus cum n. ulnari, 23: ulna, 24: radius, 25: membrana interossea antebrachii, 26: m. pronator quadratus, 27: m. extensor carpi ulnaris, 28: m. anconeus medialis, 29: a. interossea cranialis, 30: n. interosseus antebrachii, 31 n. radialis.

### 3.7. Regio Carpi

#### 3.7.1. Dorsal Approach

On the superficial plane, a large branch of the radial artery (ramus carpeus dorsalis), which lies at the craniomedial side of the antebrachium, leads towards the dorsal side of the front foot (hand) to form the arcus dorsalis (Figure 10A). This dorsal arterial arch gives off an abaxial branch running at the medial aspect of the first digit (thumb), four interdigital branches that can be found in between the first and second, the second and third, the third and fourth, and the fourth and fifth digits, and an abaxial branch running at the lateral aspect of the fifth finger. The smaller interdigital arteries that run towards the palmar side of the hand anastomose with the arcus palmaris superficialis. The insertion of the brachioradialis muscle at the distal aspect of the radius is located medial to the m. abductor digiti primi longus. Lateral to this muscle lies the common tendon of the m. extensor digitorum communis. More distally, on the dorsal side of the hand, five individual tendons, one for each finger, arise from this common tendon. Subsequently, from medial to lateral, the m. extensor digitorum secundi et tertii proprius, the m. extensor digitorum quarti et quinti proprius, and the m. extensor carpi ulnaris can be observed. Finally, the m. flexor carpi ulnaris can be seen at the lateralmost aspect of the distal antebrachium.

After the removal of the dorsal arterial arch and the deep fascia (Figure 10B), the tendons of the m. extensor digitorum quarti et quinti proprius and the m. extensor carpi ulnaris can be separated. In addition, the muscular tissue of the m. extensor carpi ulnaris is visible lateral to the tendon. The slender tendon of the m. extensor digitorum quarti et quinti proprius runs over the muscular tissue of the m. extensor digitorum secundi et tertii proprius. This tendon divides into a single tendon that inserts into the proximal phalanx of the fourth digit (m. extensor digiti quarti) and a single tendon that inserts into the middle phalanx of the fifth digit (m. extensor digiti quinti). When the tendons are followed from proximal to distal, they are covered by the ligamentum carpi dorsale that secures their positions, except for the m. extensor carpi ulnaris that runs dorsal to this structure. The thin tendon of the m. extensor digiti primi longus [m. extensor pollicis longus] can be noticed just medial to the m. extensor digitorum secundi et tertii proprius. This tendon submerges underneath the common tendon of the m. extensor digitorum communis to arise again proximal to the ligamentum carpi dorsale. During its passage underneath this ligament, it dorsally crosses the common tendon of the m. extensor carpi radialis longus et brevis. The mm. interossei between the first and second and between the second and third digits can be seen distal to this tendon. The m. adductor digiti primi [m. adductor pollicis] can additionally be seen in between the first and second digits. This muscle runs from the second and third metacarpal bones towards the proximal phalanx of the pollex. The m. abductor digiti quinti is located at the lateral side of the hand, just distal to the carpus. It has its origin on the dorsal carpal ligament and the most lateral carpal bones. Insertion is into the proximal phalanx of the fifth digit.

Finally, the ligamentum carpi dorsale is transected and several muscles have been removed (Figure 10C). The m. abductor digiti primi longus with its tendon, the tendon of the m. extensor digiti primi longus, and the m. extensor digitorum secundi et tertii proprius, with the tendon to the second digit (m. extensor digiti secundi) and the tendon to the third digit (m. extensor digiti tertii), are still present proximal to the carpus. The tendons of the m. extensor digiti secundi and of the m. extensor digiti tertii insert into the proximal phalanx of the second and third digit, respectively. As the tendons of the m. extensor digitorum communis have been retracted distally, the mm. interossei (manus) come into sight. These muscles form pairs that are present in each intermetacarpal cleft. They attach to the fringes of the metacarpophalangeal joints.

**Figure 10 vetsci-10-00164-f010:**
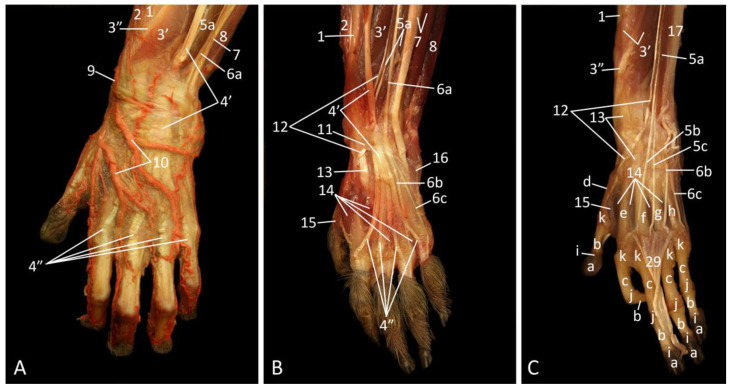
Dorsal view of the left hand; the arterial system is filled with red latex. (**A**): Superficial layer; (**B**): Deeper layer, after the removal of the arcus dorsalis and the deep fascia; (**C**): Deepest layer, after the tendons of the m. extensor digitorum communis have been transected and retracted, and the m. brachioradialis, the m. extensor carpi ulnaris, and the proximal segment of the tendon of the m. extensor digitorum quarti et quinti proprius were removed.

1: radius, 2: m. brachioradialis, 3′: m. abductor digiti primi longus, 3″: tendon of the m. abductor digiti primi longus, 4′: common tendon of the m. extensor digitorum communis, 4″: individual tendons of the m. extensor digitorum communis, 5a: m. extensor digitorum secundi et tertii proprius, 5b: tendon of the m. extensor digiti secundi, 5c: tendon of the m. extensor digiti tertii, 6a: tendon of the m. extensor digitorum quarti et quinti proprius, 6b: m. extensor digiti quarti proprius, 6c: m. extensor digiti quinti proprius, 7: m. extensor carpi ulnaris, 8: m. flexor carpi ulnaris, 9: a. radialis ramus carpeus dorsalis, 10: arcus dorsalis, 11: ligamentum carpi dorsale, 12: m. extensor digiti primi longus, 13: m. extensor carpi radialis longus et brevis, 14: mm. interossei, 15: m. adductor digiti primi, 16: m. abductor digiti quinti, 17: ulna, 18: phalanx distalis, 19: phalanx media, 20: phalanx proximalis, 21: os metacarpale primum, 22: os metacarpale secundum, 23: os metacarpale tertium, 24: os metacarpale quartum, 25: os metacarpale quintum, 26: articulatio interphalangea distalis, 27: articulatio interphalangea proximalis, 28: articulatio metacarpophalangea, 29: retracted tendons of the m. extensor digitorum communis.

#### 3.7.2. Palmar Approach

When the wrist and hand are studied by a palmar approach, it is worthwhile recapping the musculature of the antebrachium that is visible proximal to the wrist on the most superficial layer (Figure 11A). The m. flexor carpi ulnaris lies most laterally. The m. flexor digitorum superficialis is situated immediately medial to it. The tendon of the palmaris longus muscle runs superficially, over the superficial digital flexor. An analogous situation is present for the m. flexor digitorum profundus and the m. flexor carpi radialis. Both muscles are located at the medial side of the antebrachium, with the flexor carpi radialis muscle running superficially over the deep digital flexor. The m. brachioradialis, which was extensively described in the proximal segments of the thoracic limb by a lateral approach, can be seen at the medial aspect of the antebrachium. The radial artery, which provides blood to the arcus palmaris superficialis at the palmar side of the hand, equivalent to the arcus dorsalis at the dorsal side of the hand, runs between the m. flexor digitorum profundus and the m. brachioradialis. This artery and the tendons of the above-mentioned muscles are covered by the ligamentum carpi palmare at the level of the wrist. Some muscles of the hand can be identified distal to this ligament on the superficial layer. The short m. palmaris brevis lies directly subcutaneously in extension of the m. flexor carpi ulnaris. It arises from the palmar aponeurosis and is inserted into the palmar subcutis. The mm. interossei, the m. adductor digiti primi, and the m. abductor digiti quinti, which all have been studied by the dorsal approach, can also be identified at the palmar side of the hand. The m. abductor digiti quinti can be found immediately distal to the palmaris brevis muscle. More distally, at the lateropalmar side of the fifth digit, lies the m. flexor digiti quinti brevis. It inserts into the proximal phalanx of the fifth digit together with the abductor muscle of that digit. The m. abductor digiti primi brevis [m. abductor pollicis brevis] can be found at the medial side of the hand, where it arises medially from the palmar carpal ligament. It is inserted into the base of the proximal phalanx of the pollex.

The structures located at the level of the wrist and the palm of the hand can be studied by removing the superficial palmar arch and the palmar carpal ligament (Figure 11B). The various nerves immediately catch the eye. Proximal to the wrist, the ulnar nerve is situated in between the m. flexor carpi ulnaris and the m. flexor digitorum superficialis. The ramus superficialis n. ulnaris branches off when traversing the wrist, at the level of the m. abductor digiti quinti. It subsequently divides into the n. digitalis palmaris proprius V abaxialis and the n. digitalis palmaris communis IV. The latter has connection with the n. digitalis palmaris communis II et III in the middle of the palm of the hand by means of a ramus communicans. This digital nerve is a branch of the median nerve, which can be recognized proximal to the wrist as a pale skein running over the m. flexor digitorum superficialis in between the tendons of the m. flexor carpi radialis and the m. palmaris longus. When following the tendon of the palmaris longus muscle, the partially resected aponeurosis palmaris can be identified immediately distal to the wrist. Medial to this structure lies the retinaculum flexorum. At the level of the thumb, the m. flexor digiti primi brevis superficialis [m. flexor pollicis brevis pars superficialis] is located laterodistal to the m. abductor digiti primi brevis. The muscle starts on the palmar carpal ligament and is inserted into the base of the proximal phalanx of the pollex.

The retinaculum flexorum, a transverse connective tissue band that passes over the wrist and, as such, secures the positions of the tendons of the superficial and deep digital flexors, is well visible in Figure 11C. Some short muscle can bilaterally be identified. The m. flexor digiti quinti brevis that was transected and retracted can be seen at the lateral side of the hand. Since the m. abductor digiti quinti is also retracted, the deeper m. opponens digiti quinti can be observed. This muscle lies deep compared to the abductor and flexor muscles of the fifth digit. It inserts along the entire length of the fifth metacarpal bone. At the medial side, at the basis of the thumb, the m. abductor digiti primi brevis has been retracted, revealing the deeper m. opponens digiti primi [m. opponens pollicis] that runs from the palmar carpal ligament to the first metacarpal bone of the thumb. The individual tendon to the thumb of the m. flexor digitorum profundus can be seen in between the m. flexor digiti primi brevis superficialis and the m. adductor digiti primi.

The individual tendons to digits I to V of the m. flexor digitorum profundus can be studied in more detail in the next figures. In Figure 11D, they are still covered by the individual tendons of the m. flexor digitorum superficialis, except for the tendon to the thumb, since the superficial digital muscle only provides tendons to digits II–V. These four tendons of the superficial digital flexor insert into the second phalanx, whereas the tendons of the deep digital flexor insert into the distal phalanges of all five digits. After the m. opponens digiti primi and the m. flexor digiti primi brevis superficialis have been retracted, the m. flexor digiti primi brevis profundus [m. flexor pollicis brevis pars profundus] is visible. Distal to the above-mentioned retracted muscles of the thumb, a fragment of the retracted m. abductor digiti primi brevis can be seen.

The ulnar artery, which could already be identified in Figure 11D, can be followed from proximal to distal by laterally retracting the m. flexor carpi ulnaris (Figure 11E). The m. flexor digitorum profundus can be seen entirely when the superficial digital flexor is resected. The tendon of the m. flexor carpi radialis is seen medial to the common tendon of the deep digital flexor at the level of the carpus.

When this common tendon is cut proximal to the wrist and retraced, the tendon of the m. flexor carpi radialis is fully exposed (Figure 11F). The triangular shape of the m. adductor digiti primi can now be appreciated. The mm. contrahentes digitorum manus are located lateral to this muscle. Origins of the contrahentes muscles are the proximal epiphyses of the second and third metacarpal bones. Insertion is into the proximal phalanges of the fourth and fifth digits. The specific terms m. contrahens digiti quatri and m. contrahens digiti quinti can therefore be applied. Finally, the m. opponens digiti quinti lies deep compared to the abductor and flexor muscles of the fifth digit. It inserts along the entire length of the fifth metacarpal bone. The ulnar artery that is accompanied by the ulnar nerve runs at the lateral side of the distal antebrachium, where the m. pronator quadratus can be observed. Its superficial palmar arch anastomoses with the superficial palmar branch of the radial artery.

**Figure 11 vetsci-10-00164-f011:**
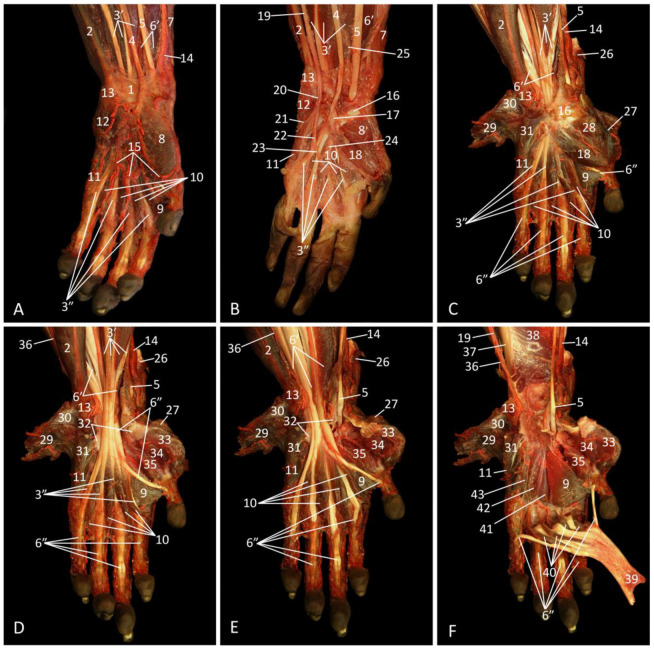
Palmar view of the left hand; the arterial system is filled with red latex, except for image B. (**A**): Superficial layer; (**B**): Second layer, after the arcus palmaris superficialis and the ligamentum carpi palmare have been removed; (**C**): Third layer, after the m. abductor digiti primi brevis, the m. abductor digiti quinti, and the m. palmaris brevis have been retracted, and after the removal of the m. palmaris longus and the superficial palmar nerves; (**D**): Fourth layer, after the m. opponens digiti primi and the m. flexor digiti primi brevis superficialis have been retracted, and after the transection of the retinaculum flexorum; (**E**): Fifth layer, after the resection of the m. flexor digitorum superficialis; (**F**): Deepest layer, after transecting and retracting the tendons of the m. flexor digitorum profundus and after the amputation of the m. flexor carpi ulnaris.

1: ligamentum carpi palmare, 2: m. flexor carpi ulnaris, 3′: m. flexor digitorum superficialis, 3″: tendons of the m. flexor digitorum superficialis, 4: tendon of the m. palmaris longus, 5: tendon of the m. flexor carpi radialis, 6′: m. flexor digitorum profundus, 6″: tendon of the m. flexor digitorum profundus, 7: m. brachioradialis, 8: m. abductor digiti primi brevis, 9: m. adductor digiti primi, 10: mm. lumbricales, 11: m. flexor digiti quinti brevis, 12: m. abductor digiti quinti, 13: m. palmaris brevis, 14: a. radialis, 15: arcus palmaris superficialis, 16: retinaculum flexorum, 17: resected aponeurosis palmaris (of the m. palmaris longus), 18: m. flexor digiti primi brevis superficialis, 19: n. ulnaris, 20: ramus superficialis n. ulnaris, 21: n. digitalis palmaris proprius V abaxialis, 22: n. digitalis palmaris communis IV, 23: ramus communicans, 24: n. digitalis palmaris communis II et III, 25: n. medianus, 26: stump of the m. brachioradialis, 27: retracted m. abductor digiti primi brevis, 28: m. opponens digiti primi, 29: retracted and cut m. flexor digiti quinti brevis, 30: retracted m. abductor digiti quinti, 31: m. opponens digiti quinti, 32: transected retinaculum flexorum, 33: retracted m. flexor digiti primi brevis superficialis, 34: retracted m. opponens digiti primi, 35: m. flexor digiti primi brevis profundus, 36: a. ulnaris, 37: radius, 38: m. pronator quadratus, 39: retracted tendons of the m. flexor digitorum profundus, 40: retracted mm. lumbricales, 41: m. contrahens digiti quatri, 42: m. contrahens digiti quinti, 43: m. opponens digiti quinti.

### 3.8. Vena Cephalica

As regards the venous circulation of the thoracic limb, it has to be mentioned that the deep venous system was not filled with latex and, therefore, not specifically examined in the present study. However, as the deep venous system largely accompanies the arterial system, the study of the arteries allows for the uncomplicated identification of the deep veins [22]. In contrast, the superficial venous system has no arterial counterpart. It mainly consists of the cephalic vein and, hence, is poorly developed in the rhesus monkey compared to man. The cephalic vein can be used for venipuncture, as is customary in companion animals, but is not preferred in the rhesus monkey.

The cephalic vein was filled with blue latex in one cadaver, in orthograde direction, at the level of the carpus. Filling veins from proximal to distal is impossible due to the presence of venous valves [23]. The topography of the cephalic vein will be described from proximal to distal, in analogy with the descriptions above of the various regions, although the blood flow in the vein is from distal tot proximal. In the cranial shoulder region, the cephalic vein emerges in between the m. cleidodeltoideus and the m. pectoralis descendens. She can subsequently be followed in distal direction cranial to the m. biceps brachii caput longum and the m. brachioradialis (Figure 12A).

Figure 12B presents a cranial view of the upper arm. The cephalic vein runs superficially over the m. biceps brachii caput longum, close to the m. brachialis. Once she has passed the flexion angle of the elbow, her trajectory can be followed over the m. brachioradialis. This is better visualized in Figure 12C. The vein can be observed at the cranial side of the antebrachium, running over the m. brachioradialis. Immediately proximal to the wrist, the injection site of the blue latex can be seen (Figure 12D). Again, the latex was injected distally, in orthograde (i.e., proximal) direction, taking into account the presence of venous valves [23].

**Figure 12 vetsci-10-00164-f012:**
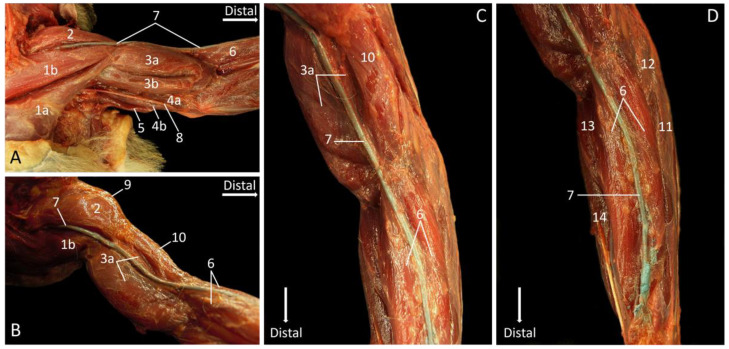
Trajectory of the left cephalic vein that was filled with blue latex. (**A**): shoulder region; (**B**): upper arm; (**C**): elbow region; (**D**): antebrachium.

1a: m. pectoralis transversus, 1b: m. pectoralis descendens, 2: m. cleidodeltoideus, 3a: m. biceps brachii caput longum, 3b: m. biceps brachii caput breve, 4a: m. triceps brachii caput mediale, 4b: m. triceps brachii caput longum, 5: m. tensor fasciae antebrachii, 6: m. brachioradialis, 7: v. cephalica, 8: n. ulnaris, 9: m. acromiodeltoideus, 10: m. brachialis, 11: m. extensor carpi radialis brevis, 12: m. extensor carpi radialis longus, 13: m. pronator teres, 14: m. flexor carpi radialis.

## 4. Discussion

Humans and non-human primates, such as rhesus monkeys, share a common ancestor that lived roughly 10 million years ago. As a result, remarkable anatomical and physiological similarities between humans and rhesus monkeys are present. These similarities have prompted researchers to investigate a wide range of biomechanical and physiological mechanisms, and to assess novel therapies in the rhesus monkey, before applying their discoveries to humans [1,3,9]. However, not only human clinicians, but also primate veterinarians will benefit from a detailed description of the anatomy of the rhesus monkey.

Although detailed anatomical knowledge of the rhesus monkey’s forelimb can provide interesting insights into the human arm, extrapolation of data must be performed prudently since some remarkable dissimilarities are present in both species. In the rhesus monkey, three rhomboideus muscles are present. These are the m. rhomboideus thoracis, the m. rhomboideus cervicis, and the m. rhomboideus capitis. The latter muscle is absent in humans [19,21]. In contrast, all three muscles can be identified in some domestic animals, such as carnivores, rabbits, and pigs [19]. Since only two rhomboid muscles are present in man, the m. rhomboideus thoracis and the m. rhomboideus cervicis are often denoted as the m. rhomboideus major and the m. rhomboideus minor, respectively [21]. The m. rhomboideus thoracis of humans is larger and thicker than the m. rhomboideus cervicis. This contrasts with the condition in domestic animals [19].

Another remarkable difference between the muscles involved in scapular motion in the rhesus monkey compared to man and domestic animals concerns the atlantoscapular muscles. As demonstrated by the presented dissections and substantiated by previously published studies [11,12,24,25], the mm. atlantoscapulares of the rhesus monkey are composed of the m. atlantoscapularis cranialis [m. atlantoscapularis anterior/inferior] and the m. atlantoscapularis caudalis [m. atlantoscapularis posterior/superior]. The m. atlantoscapularis cranialis in the rhesus monkey is considered homologous to the m. levator scapulae ventralis in cats [26] and the m. levator scapulae in humans [21,27]. However, the terms m. atlantoscapularis cranialis and m. levator scapulae (ventralis) should not be used as synonyms since these muscles seem not identical. In the rhesus monkey, the m. atlantoscapularis cranialis originates from the wing of the atlas (ala atlantis), whereas the origin of the m. levator scapulae in man is not restricted to the wing of the atlas, but also includes the transverse processes of C2–C4. In man, insertion of the m. levator scapulae is into the superior scapular angle (angulus scapularis superior) [21], whereas the m. atlantoscapularis cranialis of the rhesus monkey inserts into the distal half of the scapular spine (spina scapulae), the acromion, and the acromioclavicular joint [25]. In the cat, the origin of the m. levator scapulae ventralis includes the occiput in addition to the wing of the atlas. Insertion is restricted to the acromion [19,28].

Literature descriptions of the m. atlantoscapularis caudalis (posterior/superior) are also confounding and further complicate comparative anatomy. Our results and literature data show that in the rhesus monkey, this muscle originates in the dorsal aspect of the wing of the atlas and inserts into the dorsal margin of the shoulder blade [24,25]. The m. atlantoscapularis caudalis, and the conjoined cervical part of the serratus ventralis muscle, has previously been denoted as the m. levator scapulae dorsi in the rhesus monkey since this muscle lies in close contact with the m. serratus ventralis cervicis [24]. An analogous muscular conformation, and hence terminology, has been described and applied in the cat [28] and the human [27]. According to Barone [19], the term m. levator scapulae is a synonym of the term m. serratus ventralis cervicis in man but not in domestic mammals. This author does not mention the term m. levator scapulae dorsi, which would be more appropriate in this context. As discussed above, the term m. levator scapulae is applied in human anatomy to denote the homologous muscle of the m. atlantoscapularis cranialis.

According to Hartman and Straus (1933), the pectoral muscles of the rhesus monkey are more complex than those of man [11]. These authors state that the pectoral musculature consists of a superficial layer, denominated as the m. pectoralis major, which is composed of a capsular and a sternal part, and a deep layer composed of the musculus pectoralis minor and the m. pectoralis abdominalis. Berringer et al. (1974) make no distinction between the superficial and deep pectoral muscles but directly address the m. pectoralis major, the m. pectoralis abdominalis, and the m. pectoralis minor [12]. They also state that m. pectoralis abdominalis does not exist in man. According to the work of Bertolini and Leutert (1977) on human anatomy, the m. pectoralis major lies superficial to the m. pectoralis minor [21]. The latter muscle is a single muscle. The m. pectoralis major consists of the pars clavicularis, the pars sternocostalis, and the pars abdominalis. If the m. pectoralis major pars abdominalis is briefly referred to as the m. pectoralis abdominalis, it should be concluded that this muscle is certainly present in humans, in contrast to the statement of Berringer et al. (1974) [12]. Since the m. pectoralis major belongs to the superficial layer, the m. pectoralis abdominalis should be considered as a superficial pectoral muscle in man. This reasoning has been followed by Casteleyn and Bakker (2022), who describe the m. pectoralis superficialis in the rhesus monkey [13]. These authors state that this muscle is the larger of the pectoral muscles and is composed of a pars sternocapsularis, a pars sternalis, and a pars abdominalis (=m. pectoralis abdominalis). The smaller of the pectoral muscles is the deep pectoral muscle, denominated as m. pectoralis profundus or m. pectoralis minor. The description of the pectoral musculature of the rhesus monkey by Casteleyn and Bakker (2022) draws a clear parallel with human anatomy [13]. It can be concluded that the pectoral musculature of the rhesus monkey is not more complex than that of man.

Muscular dissimilarities in humans, rhesus monkeys, and domestic mammals are not only present in the neck and shoulder region, but also at the level of the upper arm. The rhesus monkey, and also the baboon, present both the m. coracobrachialis profundus and the m. coracobrachialis medialis [14]. In Hominidae, including man and chimpanzee, only the m. coracobrachialis medialis can be identified [14,21]. This is also the case in domestic mammals, in which the m. coracobrachialis medialis is, in short, referred to as m. coracobrachialis. The duality of the m. coracobrachialis can still, to a certain degree, be appreciated in the horse, in which this muscle consists of a long and short head [19]. Such a long and a short head are also present in the m. biceps brachii of the rhesus monkey and man [21,24]. They cannot be recognized in any of the common domestic mammals in which the m. biceps brachii is a single muscle. As a result, the term biceps is, in these species, confusing.

The anconeus muscles, including the m. anconeus lateralis [m. anconeus] and the m. anconeus medialis [m. epitrochleoanconeus], present variation in various species. The m. anconeus lateralis, which is located at the lateral side of the extension angle of the elbow, is typically present in the rhesus monkey, domestic mammals, and man. In contrast, the medially located m. epithrochleoanconeus is absent in most humans. When present, it is considered as an anatomical variation. Its presence is also exceptional in the other Hominidae. In the rhesus monkey, however, its absence can be confirmed in a minority of specimens [24]. Finally, the presence of the m. epithrochleoanconeus is standard in the cat and the rabbit [19].

Two muscles of the forearm are dissimilar in humans and rhesus monkeys. The single m. flexor digitorum profundus of the rhesus monkey is in man further complicated by the m. flexor pollicis longus [24]. This indicates an advancement in hand mobility, especially of the thumb, which is crucial in fine motor skills. In contrast, the m. palmaris longus, which is present in many primate species, including the rhesus monkey, can be absent in some Hominidae [24,29]. In humans, it can only be identified in 86% of individuals [30]. Interestingly, its absence seems not to affect grip strength [30].

Muscles that are also absent in man but present in the rhesus monkey further include the m. cutaneus trunci [m. panniculus carnosus] and the m. tensor fasciae antebrachii [m. dorsoepitrochlearis] [24]. The latter muscle can be considered as a fourth head of the m. triceps brachii. Hence, the term m. quadriceps brachii could be suggested in analogy with the m. quadriceps femoris. Such a m. quadriceps brachii is present in most non-human primates [14,22,29]. The differences between the musculature of the rhesus monkey and that of man are summarized in Table 1.

The discussion above is focused on the musculature of the rhesus monkey. Attention is paid to the differences between this species, humans, and domestic mammals. This allows for the translation of experimental data obtained in the rhesus monkey to man. Since locomotion and locomotor behavior are intensively studied in the rhesus monkey, sound knowledge of its musculoskeletal system is a prerequisite [31]. The present manuscript does not elaborate on the bones and joints since this information has previously been published [11,12,13]. The comparison with domestic mammals enables veterinary surgeons to expand their knowledge of these species to the rhesus monkey. Veterinarians who are responsible for the medical care of rhesus monkeys—in, e.g., zoos or research facilities—will benefit from the numerous photographs that simplify the identification of resemblances and differences with domestic mammal anatomy as presented in veterinary anatomical atlases. However, a comparison was not made between the rhesus monkey hand and the front foot of common domestic mammals since these structures are very species specific. Such comparative study could, however, be valuable and should be performed in the future.

Unfortunately, the vascular and nervous systems of the rhesus monkey are not comprehensively described in the existing literature [11,12,13]. The structures are primarily described in textual format with only limited availability of pictorial material. The presentation of the arteries and nerves in relation to the musculature of the rhesus monkey forelimb is certainly a strength of our publication. No discrepancies between the rhesus monkey and man regarding the occurrence and localization of the major arteries and nerves were noticed. In both species, the radial and ulnar arteries, which originate at the terminal bifurcation of the brachial artery, are prominent and important for the blood supply of the hand. In humans, a very weak a. mediana (a secondary branch of the ulnar artery), running in the interosseous space between the radius and the ulna, has been described by Barone (2000) [19] but not by Bertolini and Leutert (1977) [21]. This artery is not mentioned in previous work on the anatomy of the rhesus monkey; neither has she been noticed during the here-described dissections [11,12,13]. In contrast, the phylogenetically younger median artery plays a pivotal role in the blood supply of the front foot of domestic mammals. In carnivores, all three arteries are present; i.e., the radial, the median, and the ulnar artery. The median artery is the principal, but the other two arteries may not be overlooked. In herbivores, however, the radial and ulnar arteries are rather vestigial, with the median artery being dominant. The venous system was not studied in detail. Only the cephalic vein has been visualized. Nonetheless, the veterinary surgeon or biomedical researcher is not deprived of valuable information on the venous system, since the deep venous system consists of veins that accompany the arteries with which they have their terms in common. Furthermore, the superficial venous system of the rhesus monkey is very straightforward, and far less complex in comparison with that of humans [21].

## 5. Conclusions

This article provides topographic anatomy data on the forelimb of the rhesus monkey in order to facilitate the translation of experimental data obtained in this species to human medicine, and to assist veterinarians during, for example, wound treatment or surgery. In general, the anatomy of rhesus monkeys shows many similarities with the anatomy of humans and domestic mammals. We have described the major anatomical differences between the various species. Since these differences are not numerous and far from substantial, the rhesus monkey appears to be a valuable model for humans.

## Figures and Tables

**Table 1 vetsci-10-00164-t001:** Differences observed between the musculature of the front limb of the rhesus monkey and that of the human arm.

Rhesus Monkey	Man
m. rhomboideus thoracis, cervicis et capitis	m. rhomboideus major et minor, capitis absent
m. atlantoscapularis cranialis (anterior/inferior)	m. levator scapulae
m. atlantoscapularis caudalis (posterior/superior) + m. serratus ventralis cervicis	m. levator scapulae dorsi
m. pectoralis superficialis (pars sternocapsularis, pars sternalis, pars abdominalis = m. pectoralis abdominalis)	m. pectoralis major (pars clavicularis, pars sternocostalis, pars abdominalis = m. pectoralis abdominalis)
m. pectoralis profundus	m. pectoralis minor
m. coracobrachialis profundus et medialis	m. coracobrachialis profundus absent
m. anconeus medialis present in most rhesus monkeys	m. epitrochleoanconeus absent in most humans
m. flexor digitorum profundus: single	m. flexor digitorum profundus + m. flexor pollicis longus
m. palmaris longus	m. palmaris longus sometimes absent
m. cutaneus trunci (m. panniculus carnosus)	absent
m. tensor fasciae antebrachii (m. dorsoepitrochlearis)	absent

## Data Availability

Not applicable.

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
