# Peer review of "Topographical Anatomy of the Rhesus Monkey (*Macaca mulatta*)—Part I: Thoracic Limb"

_vetsci, 2023, doi:10.3390/vetsci10020164_

Round 1
Reviewer 1 Report
The manuscript written by Christophe Casteleyn et al. is an excellent study of the anatomy of the thoracic limb of the rhesus monkey. I am impressed by this comprehensive, detailed description and the beautiful photographs. The work is of great importance to researchers and veterinarians. Well done! I have included minor suggestions below:
- keywords should not be a repetition of the words in the title
- references - there should be no number 32 in line 1107
- Consider unifying all anatomical names to English with Latin in the bracket, e.g. line 133 - name to cranial cervical part
- I think the article will benefit from the magnification of the figures to be more readable
- Because I mainly focus on angiology, I would be happy to read at least a short paragraph discussing this topic. The authors limited themselves to stating, "Comparison was, however, not made between the rhesus monkey hand and the front foot of the common domestic mammals since these structures are very species specific." This is true, but an indication of the main differences between the rhesus monkey and domestic animals would have been warranted.
Author Response
On behalf of the co-authors, I would like to thank you for your constructive feedback on our manuscript Topographical anatomy of the rhesus monkey (Macaca mulatta) – Part I: thoracic limb". We have incorporated most of your suggestions to improve the manuscript’s overall quality.
Keywords should not be a repetition of the words in the title
This manuscript on the anatomy of the thoracic limb is the first part of a series of two on the anatomy of the limbs of the rhesus monkey. The second manuscript is on the pelvic limb. That manuscript has already been reviewed and revised without any comment on the key words that read as follows: “anatomy; topographical anatomy; rhesus monkey; pelvic limb”. For reasons of uniformity, we would like to keep the key words of the first manuscript as they are, i.e., “anatomy; topographical anatomy; rhesus monkey; thoracic limb”.
References - there should be no number 32 in line 1107
You are correct. We have removed this reference (that is not complete – it seems that something went wrong while editing the manuscript).
Consider unifying all anatomical names to English with Latin in the bracket, e.g. line 133 - name to cranial cervical part
Currently, the Latin term is used when a structure is mentioned for the first time. Further on in the text we use the English term, unless the English term is rather awkward or does not enhance the readability of the text. This was also done in the other manuscript.
I think the article will benefit from the magnification of the figures to be more readable
We are bound to the dimensions of the journal and do not want to split up the multipanel figures because these show the progression of the dissections. However, since this manuscript will be available as online open access manuscript, the readers will be able to digitally enlarge the figures.
Because I mainly focus on angiology, I would be happy to read at least a short paragraph discussing this topic. The authors limited themselves to stating, "Comparison was, however, not made between the rhesus monkey hand and the front foot of the common domestic mammals since these structures are very species specific." This is true, but an indication of the main differences between the rhesus monkey and domestic animals would have been warranted.
Comparative anatomy between the rhesus monkey and the domestic mammals is indeed very interesting and we share the reviewer’s opinion that a short paragraph could be spent on the comparative angiology. We have included a paragraph in the Discussion in which we present the difference between the arterial blood supply of the rhesus monkey, man and domestic mammals. This paragraph can be found in lines 1036-1051. We would, however, like to mention that this paragraph only gives the essentials and not the details since we are currently performing anatomical studies on the ‘hand’ and ‘foot’ of the rhesus monkey. This is the reason why the ‘hand’ is not described in its finest details in the present manuscript.

Reviewer 2 Report
Firrs of all, congratulations for the quality of the dissections presented in this manuscript. They really made possible to observe the anatomical contents of the toracic limb in a topographical manner.
The use of photographs to ilustrate the manuscript and the legends in Latin made easier to reach different readers around the world.
Concerning the miswritings, there is one extra "t" in line 864.
I am sure that this manuscript will help researchers to perform better studies using rhesus monkeys consulting this very detailed anatomy manuscript.
Author Response
On behalf of the co-authors, I would like to thank you for your constructive feedback on our manuscript Topographical anatomy of the rhesus monkey (Macaca mulatta) – Part I: thoracic limb". We have incorporated your suggestion to improve the manuscript’s overall quality.
Concerning the miswritings, there is one extra "t" in line 864.
Thank you for noticing this. We have corrected the error.

Reviewer 3 Report
The paper deals with the classical anatomy of the thoracic limb of the rhesus monkey. The authors used an anatomical preparation method and intravenous injections of latex with dye. The paper fills an important gap in knowledge regarding the thoracic limb. The subject is treated comprehensively, i.e. muscles, vessels and nerves are described. The work has important cognitive as well as practical value from the point of view of both veterinary medicine and experimental medicine. In addition, the paper addresses some issues of anatomical nomenclature. The work has an appropriate layout, the illustrations are of adequate quality and the literature is adequate. The discussion is written in an interesting way capturing the differences between the ape species studied, humans and domestic animals.
[64] Regio scapularis is a topographical theme. The term cingulum membrii thoracici (shoulder girdle) should be used for the sentence
[172] it is better to use Th rather than T as an abbreviation for thoracic vertebrae
[209] The Nomina Anatomica Veterinaria does not distinguish the m. pectoralis abdominalis either as an independent muscle or as part of the m. pectoralis profundus (https://www.wava-amav.org/wava-documents.html). The situation is similar in animal anatomy textbooks (e.g. Dyce, Sack and Wensing; Konig, Liebich; Ellenberger/Baum). My personal opinion is that m. pectoralis profundus has a part that can be referred to as m. pectoralis profundus pars abdominalis or possibly m. pectoralis abdominis. As the paper is an anatomical study please further characterise the m. pectoralis abdominalis described by the authors and refer to this structure in the discussion.
[281] suggests a change from T1 to Th1
[289] suggests a change from T1 to Th1
[309] suggests a change from T1 to Th1
Author Response
On behalf of the co-authors, I would like to thank you for your constructive feedback on our manuscript Topographical anatomy of the rhesus monkey (Macaca mulatta) – Part I: thoracic limb". We have incorporated your suggestions to improve the manuscript’s overall quality.
[64] Regio scapularis is a topographical theme. The term cingulum membrii thoracici (shoulder girdle) should be used for the sentence
Thank you for this remark. Regio scapularis was changed to cingulum membri thoracici.
[172] it is better to use Th rather than T as an abbreviation for thoracic vertebrae
We have changed T into Th.
[209] The Nomina Anatomica Veterinaria does not distinguish the m. pectoralis abdominalis either as an independent muscle or as part of the m. pectoralis profundus (https://www.wava-amav.org/wava-documents.html). The situation is similar in animal anatomy textbooks (e.g. Dyce, Sack and Wensing; Konig, Liebich; Ellenberger/Baum). My personal opinion is that m. pectoralis profundus has a part that can be referred to as m. pectoralis profundus pars abdominalis or possibly m. pectoralis abdominis. As the paper is an anatomical study please further characterise the m. pectoralis abdominalis described by the authors and refer to this structure in the discussion.
The N.A.V. does not mention the term m. pectoralis abdominalis. We can, however, find that term in several works on the anatomy of the rhesus monkey and man. We have added a paragraph on this topic in the Discussion. It reads as follows: According to Hartman and Straus (1933) the pectoral muscles of the rhesus monkey are more complex than those of man. These authors state that the pectoral musculature consists of a superficial layer, denominated as the m. pectoralis major, which is composed of a capsular and a sternal part, and a deep layer composed of the musculus pectoralis minor and the m. pectoralis abdominalis. Berringer et al. (1974) make no distinction between the superficial and deep pectoral muscles but directly address the m. pectoralis major, the m. pectoralis abdominalis and the m. pectoralis minor. They also state that m. pectoralis abdominalis does not exist in man. According to the work of Bertolini and Leutert (1977) on human anatomy, the m. pectoralis major lies superficial to the m. pectoralis minor. The latter muscle is a single muscle. The m. pectoralis major consists of the pars clavicularis, the pars sternocostalis and the pars abdominalis. If the m. pectoralis major pars abdominalis is briefly referred to as the m. pectoralis abdominalis, it should be concluded that this muscle is certainly present in humans, in contrast to the statement of Berringer et al. (1974). Since the m. pectoralis major belongs to the superficial layer, the m. pectoralis abdominalis should be considered as a superficial pectoral muscle in man. This reasoning has been followed by Casteleyn and Bakker (2022) who describe the m. pectoralis superficialis in the rhesus monkey. These authors state that this muscle is the larger of the pectoral muscles and is composed of a pars sternocapsularis, a pars sternalis and a pars abdominalis (= m. pectoralis abdominalis). The smaller of the pectoral muscles is the deep pectoral muscle, denominated as m. pectoralis profundus or m. pectoralis minor. The description of the pectoral musculature of the rhesus monkey by Casteleyn and Bakker (2022) draws a clear parallel with human anatomy. It can be concluded that the pectoral musculature of the rhesus monkey is not more complex than that of man.
In analogy to what was asked by the reviewers of our manuscript on the pelvic limb, we have included Table 1 that presents the main differences between the musculature of the front foot of the rhesus monkey and that of man.
[281] suggests a change from T1 to Th1
We have changed T into Th.
[289] suggests a change from T1 to Th1
We have changed T into Th.
[309] suggests a change from T1 to Th1
We have changed T into Th.
In the legend of Fig. 4, we have added “(fifth cervical nerve)” after C5 etc. to make clear that not the vertebrae but the spinal nerves are indicated.
